# Obesity and Its Multiple Clinical Implications between Inflammatory States and Gut Microbiotic Alterations

**DOI:** 10.3390/diseases11010007

**Published:** 2022-12-29

**Authors:** Walter Milano, Francesca Carizzone, Mariagabriella Foia, Magda Marchese, Mariafrancesca Milano, Biancamaria Saetta, Anna Capasso

**Affiliations:** 1UOSD Eating Disorder Unit, Mental Health Department, ASL Napoli 2 Nord, 80027 Napoli, Italy; 2Mental Health Department ASL Napoli 2 Nord, 80027 Napoli, Italy; 3Clinical Pathology Services, Santa Maria Delle Grazie Hospital Pozzuoli, Asl Napoli 2 Nord, 80027 Napoli, Italy; 4Department of Pharmacy, University of Salerno, Fisciano, 84084 Salerno, Italy

**Keywords:** obesity, inflammation, microbiota, insulin resistance, metabolic syndrome, type 2 diabetes

## Abstract

Obesity is a chronic multifactorial disease that has become a serious health problem and is currently widespread over the world. It is, in fact, strongly associated with many other conditions, including insulin resistance, type 2 diabetes, cardiovascular and neurodegenerative diseases, the onset of different types of malignant tumors and alterations in reproductive function. According to the literature, obesity is characterized by a state of low-grade chronic inflammation, with a substantial increase in immune cells, specifically macrophage infiltrates in the adipose tissue which, in turn, secrete a succession of pro-inflammatory mediators. Furthermore, recent studies on microbiota have postulated new possible mechanisms of interaction between obesity and unbalanced nutrition with inflammation. This intestinal “superorganism” complex seems to influence not only the metabolic balance of the host but also the immune response, favoring a state of systemic inflammation and insulin resistance. This review summarizes the major evidence on the interactions between the gut microbiota, energetic metabolism and host immune system, all leading to a convergence of the fields of immunology, nutrients physiology and microbiota in the context of obesity and its possible clinical complications. Finally, possible therapeutic approaches aiming to rebalance the intestinal microbial ecosystem are evaluated to improve the alteration of inflammatory and metabolic states in obesity and related diseases.

## 1. Introduction

In recent decades, globally, there has been a broad explosion of obesity, and the prevalence of overweight and obesity has doubled since 1980, to the point that almost a third of the world’s population is now classified as overweight or obese. Obesity rates have increased for all ages and in both sexes, regardless of geographic location, ethnicity or socioeconomic status [1]. Rapid economic growth, with the consequent transition from a traditional and frugal lifestyle to a modern one, which is mainly sedentary and characterized by the easy and abundant availability of food, especially industrialized and rich in fats and carbohydrates, leads to significant changes in dietary patterns and preferences [1].

Obesity is currently considered a chronic multifactorial disease, resulting from the interaction between genetic load, environment and endocrine function, and factors including behavioral patterns, such as lifestyle and food choices, can differ widely from subject to subject [2].

From a clinical descriptive point of view, it is defined as an excessive accumulation of body fat and is substantiated by a body mass index (BMI) greater than 30 kg/m^2^. Given its ubiquitous spread, it is rising as one of the major public health problems. Comorbidities associated with obesity are insulin resistance, type 2 diabetes, atherosclerosis, cardiovascular diseases, hypertension, osteoarthritis and various types of cancer, significantly reducing life expectancy. In fact, obesity ranks fifth among the causes of death worldwide [2], and it is estimated to be the leading cause of death of approximately 3.4 million people every year. Due to the severity and complexity of its associated complications, it constitutes an enormous financial burden, not only for the affected but also for health systems and society in general [1,2,3,4,5].

Analyzing the economic, social and health costs linked to the body weight of the population of 52 countries in the OECD area, including 28 European Union and G20 countries, the evidence shows that, on average, states spend 8.4% of the health system budget to treat diseases related to excessive weight.

From an epidemiological point of view, it is estimated that by 2030, approximately 38% of the world’s adult population will be overweight and another 20% obese [6]. The data on the prevalence of obesity in the United States is as significant as it is worrying. The National Health and Nutrition survey Examination Survey (NHANES), performed in the USA, reported that from 1988 to 1994, from 1999 to 2000 and from 2017 to 2018 the overall prevalence of obesity, adjusted for age, increased progressively from 22.9 to 30.5 to 42.4% [7]. The prevalence of obesity was similar in adult males and females in 2017–2018 [8]. The trends are also similar globally. Worldwide obesity was estimated at around 604 million adults and 108 million children in 2015 [9]. Since 1980, the prevalence has doubled in more than 70 countries and shows a continuously increasing trend in many countries, with a similar incidence between males and females and for all age groups, albeit with higher peaks in early adulthood; in fact, the most evident increase in obesity occurred in males living in low–middle-income countries, aged between 25 and 29 years, from 1980 to 2015, passing from 11.1 to 38.3%. These are countries where the socioeconomic difficulties lead to the major consumption of low-quality food that has a major energetic impact.

Increasing trends also include the most severe forms of obesity: the age-adjusted prevalence of class III obesity (BMI ≥ 40 kg/m^2^) increased from 5.7% in 2007 to 9.2% in 2018.

The highest prevalence of obesity has been observed in some Pacific islands, with rates close to 80%. Conversely, the lowest obesity rate, in less than 4% of the population, was reported in India and Vietnam [10]. 

In Europe, in general, the incidence of obesity is high with significantly uneven prevalence data that have been reported between the different regions of the continent. The lowest rate was observed in Tajikistan [13.5%] and the highest in Andorra and Turkey (29.4%) [11,12].

Awareness of the negative impact of obesity on health is now well established, so much so that the American Medical Association (AMA), the largest medical organization in the United States, recently released a communication defining obesity as a disease on its own [13], with a strong impact on general physical conditions; although, it should be emphasized, that the development of metabolic and cardiovascular complications are not inevitable in obese individuals [14]. It is therefore considered of fundamental importance, both from a clinical and epidemiological point of view, to identify obese individuals with a high risk of diseases associated with obesity as early as possible, since this would allow a lower general clinical impact and a more effective use of economic resources [15].

Both from a biological and cognitive point of view, in humans the regulation of food intake is based on an intricate feedback system. Signals of hunger and satiety originate both in the brain and in peripheral tissues and organs through two complementary pathways that include the homeostatic and hedonic pathways [1]. Among the different hormones involved in this fine regulation of caloric intake, insulin plays a central role. 

In recent years, it has become increasingly evident how insulin resistance has a direct role in developing metabolic and cardiovascular complications (i.e., the onset of metabolic syndrome (MetS) [16] and not only diabetes type 2.

Although the clinical features contributing to the definition of a metabolic syndrome may vary, there is general agreement on the crucial involvement of abdominal obesity and insulin resistance in causing lipid alterations, hyperglycemia and hypertension, which commonly identify the syndrome and make it a major risk factor for the onset of cardio metabolic disease [16]. Metabolic syndrome should be considered as a clinical diagnosis guided by a complex combination of factors, including impaired fat storage, insulin action and pro-inflammatory factors [17].

Overall, MetS describes a syndromic cluster or constellation of cardiovascular and diabetes risk factors within a single individual. This includes abdominal adiposity (i.e., an increase in waist circumference), hypertension, reduced lipoprotein (HDL) levels, increased triglycerides and glucose intolerance. Three of these five criteria that exceed the cut-off values determine the diagnosis of MetS [16].

The excessive intake of poor-quality nutrients, sedentary lifestyles, environmental toxins and endocrine disruptors (e.g., digoxin and bisphenol A) induce both the onset of obesity and MetS and the related internal complications [5]. Several drugs can also promote excess weight and the onset of MetS, such as thiazide diuretics, beta blockers, niacin, thiazolidinediones in type 2 diabetes mellitus (T2DM), oral contraceptives, protease inhibitors, several atypical antipsychotics, tricyclic antidepressants and monoamine oxidase inhibitors, some antiepileptic drugs and glucocorticoids [17,18,19].

## 2. Inflammation

Low-grade inflammation is heavily involved in the link between obesity and the progression of associated conditions, such as insulin resistance and type 2 diabetes [20,21].

Over the past decade, the search for a potential common mechanism underlying the pathogenesis of obesity-associated diseases has revealed a close relationship between nutrient excess and imbalances in the cellular and molecular mediators of immunity and inflammation [1].

Inflammation is a defense mechanism, typical of innate immunity, which, in the event of infections and injuries, aims to locate and eliminate the harmful agent that underlies it; it also removes damaged components, repairs damages to the tissues and restores the normal functionality of the organism through defensive cells, promoting healing. Inflammation, therefore, is a multisystem response to harmful stimuli, with the aim of bringing the system back to a level of equilibrium.

The inflammatory response triggered by nutrient excess in obesity involves many components of the classic inflammatory response to pathogens. This includes systemic increases in circulating inflammatory cytokines and in acute-phase proteins, such as the recruitment of leukocytes into inflamed tissues, activation of tissue leukocytes and the processing of tissue reparative responses, such as fibrosis and C-reactive protein (CRP) [22,23]. C-reactive protein (CRP) is an index of inflammation; as such, its blood concentration increases in the case of various types of inflammatory processes [23,24,25].

When there is a positive energy balance, excess energy in the form of triglycerides accumulates in the subcutaneous adipose tissue. This leads to hyperplasia of the subcutaneous adipose tissue, i.e., proliferation and differentiation of pre-adipocytes. When the subcutaneous adipose tissue is no longer able to store excess energy correctly or the storage threshold has been exceeded, visceral fat deposits increase; this kind of tissue, having a lower adipogenic capacity, grows mainly due to the fact of hypertrophy, i.e., by the increasing size of adipocytes [2].

Deposits of adipocytes are found throughout the body, albeit in varying amounts, in many organs, including the heart and kidney, bone marrow, lungs, and the adventitia of major blood vessels [24]. Evidence shows that those fat deposits, in the case of high-calorie diets, go through the same inflammatory process as the subcutaneous and visceral adipose tissues [25,26].

In obesity, the increase in the adipocytes, due to the fact of hypertrophy, determines a dysregulation of the generalized adipose tissue, which involves a remodeling of the structure and, subsequently, a condition of inflammation, with both local and systematic repercussions [2].

The first evidence to support the connection between inflammation and metabolic disorders arose at the end of the last century. Hotamisligil and Spiegelman [27,28] highlighted the presence of TNF-alpha (a pro-inflammatory cytokine produced mainly by macrophages that stimulates the inflammatory reaction in the acute phase) in adipose tissue and its direct role in promoting the condition of insulin resistance in mice. Further human studies have confirmed the presence of higher levels of CRP, IL-6 and IL-18 in obese people than those of normal weight [29,30]. Furthermore, prospective and cohort studies indicate that the plasma levels of CRP and adiponectin modulate the risk of developing type 2 diabetes [31].

Obesity, metabolic syndrome and type 2 diabetes are associated with an inflammatory pattern characterized by the overexpression of pro-inflammatory cytokines and a reduced production of adiponectin, a protective factor, suggesting a close relationship between these biomarkers, metabolic disorders and cardiovascular risk [32]. It is hypothesized that visceral adiposity is the starting condition through the release of pro-inflammatory cytokines, which can promote insulin resistance, one of the main factors favoring a state of hyperglycemia in predisposed people [33].

Several studies, both on experimental animals and in humans, have shown that in the early stages of the expansion of the adipose tissue, linked to hypertrophy of the adipocytes, areas of hypoxia develop, that is, poorly oxygenated adipose tissue. It has been observed that these areas of hypertrophic adipose tissue, in the case of hypoxic conditions, secrete pro-inflammatory adipokines, such as the macrophage migration inhibitory factor (MIF), extracellular matrix proteinases (MMP2 and MMP9), IL-6, plasminogen activator inhibitor-1 (PAI-1), vascular endothelial growth factor (VEGF) and leptin [17,18]. At the same time, the lack of oxygen causes the death of the more peripheral adipocytes, triggering an increase in the inflammatory reaction [2].

Inflammation, therefore, in the first phase of nutritional excess, can play a protective role in the adaptive responses to overnutrition, favoring tissue remodeling, increasing the expansion of adipose tissue.

Physiologically, the expansion of adipose tissue requires adipogenesis and angiogenesis, both creating space and support for the nutritional needs of the newly formed adipocytes. The endothelium supports them by enabling the exchange of nutrients from the blood and lymph. Angiogenesis, in general, facilitates the access to oxygen, among others, to allow for storage of new lipids and their mobilization whenever lipolysis occurs. This expansion is also essential for preventing ectopic deposition from occurring in tissues, such as liver or muscle [34,35]. 

Immune cells are able to cross the endothelial barrier to enter the adipose tissue; hence, during the early stages of obesity, inflammation plays an important role in supporting this adaptive response. However, if the inflammation persists, angiogenesis and adipose tissue expansion will eventually favor the onset of insulin resistance as well as fibrosis, adipocyte dysfunction and even cell death. A review by Philipp Scherer and colleagues accurately describes the interactions between inflammation, angiogenesis and fibrosis in the context of adipose tissue expansion [36].

The tonic and constant low-grade activation of the innate immune system, induced by obesity, can affect metabolic homeostasis overtime. The nature of inflammation induced by obesity is peculiar compared to other inflammatory paradigms (e.g., infections and autoimmune diseases) in several key aspects; it can involve several complex metabolic systems and have multi-organ effects and is therefore to be considered one of the key linkages between obesity, insulin resistance, metabolic syndrome, T2DM, nonalcoholic steatohepatitis (Nash) and cardiovascular disease [37,38]. 

In relation to all of these characteristics, in recent decades, it has been hypothesized as a low-grade chronic “metabolic” inflammation (local and systemic), also called “metaflammation” [39].

Naturally, these phenomena must also be seen in the context of a single individual, i.e., the presence of overexpressed genes associated with obesity and metabolic diseases [40].

### 2.1. Activation of the Innate Immune System in Obesity

In the adipose deposits, there are normally different types of immune cells that, together, monitor and maintain the integrity and hormonal sensitivity of the adipocytes. All metabolic tissues contain resident populations of leukocytes, indicating that the immune system is ready to respond to nutrient-derived signals [41]. These immunocompetent cells release a cascade of cytokines that regulate different types of immune cells in a coordinated way, such as eosinophils, mast cells and, especially, macrophages [5].

Macrophages are remarkably plastic cells that can take on a range of different phenotypes to adapt to different tissue microenvironments. Consequently, macrophages can exhibit pro- or anti-inflammatory phenotypes and are routinely classified into M1 phenotype (classically activated) and M2 phenotype (alternatively activated). According to this classification, macrophages acquire the M1 phenotype upon stimulation with interferon gamma (IFN-γ) alone or in combination with TLR ligands (e.g., lipopolysaccharide (LPS)), while macrophages acquire the M2 phenotype after exposure to IL-4 and IL-13. M1 macrophages secrete high levels of pro-inflammatory cytokines (e.g., tumor necrosis factor (TNF-α), IL-6 and IL-1β). On the contrary, M2 macrophages release IL-10 and other protective cytokines essential for the resolution of the inflammatory response and contribute to the maintenance of a balanced condition between pro- and anti-inflammation systems and insulin sensitivity in the adipocytes [23,41].

In states of overnutrition and obesity, the phenomenon of “phenotypic change” occurs (also called “phenotypic switch”); it is the change in the polarization state of macrophages, from the form with a prevalent anti-inflammatory action in the M2 state (i.e., the predominant form during the energy balance in equilibrium) to a pro-inflammatory M1 state, which is predominant in conditions of obesity [42].

In the mechanism underlying inflammation, several inputs contribute to metabolic dysfunction, such as the increase in circulating cytokines as well as the decrease in protective factors (e.g., adiponectin) and the interactions between inflammatory and metabolic cells. For example, the direct and paracrine signals of activated macrophages, or M1, can compromise insulin signaling on target tissues as well as the adipogenesis of adipocytes; conversely, nonactivated macrophages, or M2, do not determine these conditions [23,43]. This evidence leads us to evaluate immune activation not only from the classic pro-inflammatory paradigm. The extent of macrophage infiltration into adipose tissue (ATM) is also influenced by the different range of activation states, which are dependent on local stimuli [44]. After stimulation by lipopolysaccharide (LPS) and interferon gamma (IFN-γ), macrophages develop the classic state of pro-inflammatory activation (M1), which generates bactericidal or Th1 responses, which are typically associated with obesity. 

Conversely, Th2 cytokines, such as IL-4 and IL-13, create an activated macrophage (M2) state that promotes fibrotic response and attenuation of the classic pro-inflammatory and activation pathways, which normally depend on the nuclear kappa-light-chain-enhancer of activated B cells (NF-κB).

Macrophages that infiltrate adipose tissue can therefore have a variety of intermediate levels along the M1/M2 spectrum, depending on the location of fat deposits and the levels of nutritional status [45]; the increase in adiposity determines a shift towards the inflammatory profile, where the classic M1 pro-inflammatory signals predominate [23,45]. 

In adipose tissue, the distinction between the polarization of M1 and M2 macrophages can be monitored by evaluating the expression of selected markers. Macrophages showing the M1 phenotype are characterized by the expression of F4/80, CD11c and iNOS, while macrophages showing the M2 phenotype are characterized by the expression of F4/80, CD301 and Arg1 [37].

Macrophages therefore represent the effectors of a complex immune system triggered by overeating. Macrophages undergo significant changes during obesity; the overall number of macrophages increases due in large part to the recruitment of M1-polarized macrophages; in addition, they secrete cytokines, such as TNF-α, expressing a markedly pro-inflammatory phenotype. The increase in the number of macrophages in this state results in an increase in the ratio of M1 to M2 macrophages, which is a hallmark of the inflammation of the adipose tissue that accompanies obesity and is associated with the development of insulin resistance and metabolic disease [45]. Consequently, the presence of macrophages in the adipose tissue can increase from 10% to approximately 40% of the total number of cells present, especially the M1 population. The interaction between the activated macrophages M1 and the adipocytes is therefore a relevant factor in the development of obesity and insulin resistance. Conversely, weight loss reduces the infiltration of macrophages and the expression of factors correlated with inflammation of the adipose tissue [2].

Tumor necrosis factor (TNF), which is mainly expressed and secreted by macrophages in the M1 state in adipose tissue, is one of the first cytokines found to be increased in adipose and circulating tissue in people with T2DM and obesity [27,46]. TNF expression in adipose tissue is inversely related to insulin sensitivity in obese people without T2DM compared to healthy and lean controls. Weight loss in obese subjects therefore reduces TNF expression in adipose tissue and improves insulin sensitivity, suggesting that TNF plays a substantial role in inflammation and insulin sensitivity in obese subjects [39,47].

Furthermore, within the pancreatic islets, the production of TNF by macrophages can promote a dysfunction of the beta cells and can directly lead to insulin resistance [48]. TNF can play a crucial role in the development of insulin resistance in different tissue types, with high glucometabolic relevance.

Although macrophages are the cell line predominantly involved in inducing the inflammatory condition of adipose tissue, other cell lines also contribute significantly through a complex system of interactions between different immune cells. In addition to macrophages, innate lymphoid cells (ILCs) can be found in adipose tissue, which play the role of “immune sentinels” and are capable of “modeling” subsequent immune responses. Based on the production and the class of cytokines produced, three cell subpopulations are identified: ILC1, ILC2 and ILC3, which operate in total synergy with natural killer cells.

Natural killer (NK) cells and ILC1s appear to contribute to the obesity phenotype by promoting a pro-inflammatory environment. Natural killer cells are white blood cells involved in both innate immune responses and acquired immunity and can produce various cytokines, such as IL-1, TNF-alpha or interferon-gamma (IFN-γ), which can influence acquired immunity, promoting the differentiation of T-helper type 1 (TH1) cells and inhibiting that of T-helper type 2 (TH2) cells [21,23].

The TH1 response is cytotoxically oriented towards viruses and bacteria. It is supported by IFN-γ, which activates the production of free radicals and nitric oxide (NO), especially by macrophages, and inhibits the TH2 response, and by IL-12.

The TH2 response is antibody-oriented and supported by IL-4 (which activates B lymphocytes and Ig E production), IL-5 (which recruits eosinophils in the presence of parasites), IL-13 and IL-10 (anti-inflammatory cytokine; blocks IL-3, IL-5 and IL-12; IFN-γ production; and TH1 response) but is pro-inflammatory against allergic processes [21,23].

Under conditions of energy balance, ILCs maintain a reduced number of macrophages in the adipose tissue to favor the metabolic homeostasis of the adipose tissue [49]. When the energy balance is altered, in the presence of a diet rich in fats, an infiltration of pro-inflammatory macrophages in the adipose tissue is triggered, and the excess of dietary fats favors not only the increase in the number of NK cells and the production of pro-inflammatory TNF in the adipose tissue but also the proliferation of ILC1 in the adipose tissue, which further favors the development of a pro-inflammatory environment through the secretion of IFN-γ [39,50]. The maintenance of metabolic homeostasis, therefore, requires a balanced immune response and an integrated network between the various types of immunocompetent cells [23].

The inflammatory condition of the adipose tissue, as we have already mentioned, favors the reduction of insulin sensitivity and, subsequently, a real insulin resistance.

Insulin resistance is characterized by the maintenance of high plasma insulin concentrations but with a decrease in the ability of cells to respond to insulin action. Insulin is a polypeptide hormone with an important regulatory function on metabolism. It binds to its transmembrane receptors which are present in the muscle, liver, adipose tissue and the central nervous system (CNS). Its action is characterized by the promotion of a sense of satiety, an increase in energy expenditure and the regulation of the action of leptin.

Insulin resistance and, consequently, the reduced insulin response capacity of the target cells, subjects the tissues to an inadequate lipogenic action, which promotes obesity, hepatic production of triglycerides and the release of very low-density lipoproteins (VLDLs), resulting in the development of dyslipidemia. Insulin resistance also increases the risk of atherosclerosis through the addition of cardiovascular risk factors, endothelial alterations and inflammatory and coagulation processes [2]. Therefore, insulin resistance is an important factor that initiates some of the characteristics of metabolic syndrome [51].

A recent study suggests that obesity-related insulin resistance chronologically precedes the pro-inflammatory macrophages’ infiltration of tissues [52]. Firstly, it has been shown that insulin resistance is associated with the increased presence of some cytokines, namely, chemokines, that are able to attract immune cells to metabolically active tissues. Among these, the monocyte chemoattractant protein-1 (MCP-1) is a potent attraction factor for monocytes [52,53] and facilitates the infiltration and differentiation of monocytes into pro-inflammatory M1 macrophages. Second, insulin resistance in wild-type mice precedes macrophage accumulation during diet-induced obesity (DIO). Third, the adipose tissue of obese patients with insulin resistance has a lower mechanistic target of rapamycin complex 2 (mTORC2) signaling, secondary to high MCP1 expression and the higher presence of macrophages [52]; mTORC2 plays an important role in regulating energy balance and weight and appears to be regulated by insulin, growth factors and the quantity of nutrients [54].

In turn, pro-inflammatory cytokines, such as TNF-α and IL-6, hyper-expressed by activated M1 macrophages, favor the increase in lipid deposition because hyperinsulinemia decreases the sensitivity of the insulin receptor at the CNS level, reducing the feeling of satiety, instead increasing the feeling of hyperphagia and, therefore, facilitating a positive energy balance which, in turn, favors the hypertrophy of the adipocytes, perpetuating the cycle of obesity–lipoinflammation–increased appetite [2,55].

This experimental evidence could explain the temporal sequence of the association between nutrient excess, obesity and insulin resistance. Insulin resistance could be the consequence of obesity and the cause of infiltration by macrophages in the adipose tissue which, in turn, amplify the possibility of favoring the evolution into diabetes.

Additionally, T cells in obese fat tissue produce more pro-inflammatory mediators than lean controls. The levels of interferon gamma (IFN- γ), produced by T-helper (Th) 1 cells in human visceral adipose tissue, correlate positively with the systemic inflammatory state but are not associated with insulin resistance, while anti-inflammatory Th2 cells exhibit a protective connotation with respect to insulin resistance. Experimental data on obese mice lacking IFN-γ show adipose tissue with lower inflammation and better glucose control [56].

As a final consequence, inflammation of the adipose tissue and insulin resistance can promote not only stability but also increase the condition of obesity; they support hepatic liponeogenesis as a consequence of the reduced use of carbohydrates in peripheral tissues [57]. Moreover, insulin resistance, favored by inflammation, decreases the feeling of satiety, inducing excessive food intake [2,55].

Therefore, it could be said that inflammation is not only a consequence of obesity but could also be involved in its maintenance, favoring progressive weight gain and, thus, generating a positive feedback loop.

Weight loss, on the other hand, tends to reduce inflammation of the adipose tissue, regardless of the way it is performed, i.e., diet and/or physical activity or bariatric surgery. Weight loss of at least 5% results in a significant decrease in the circulatory levels of inflammatory mediators in the obese with no other chronic comorbidities. A meta-analysis of studies reporting data on obese people who experienced weight loss (through lifestyle or surgery), performed between 1966 and 2006, showed that for every kilogram of weight loss, the CPR levels were reduced by 0.13 mg/L [58,59].

Obesity and MetS are closely associated with a chronic low-grade inflammatory process, clinically characterized by a more or less modest but harmful increase in serum CRP [60,61,62]. 

### 2.2. Microbiota

There is unquestionably a “cross dialogue” in the communication mechanisms between signaling pathways linked to inflammation and the microbiota in favoring the onset of obesity and metabolic syndrome following food excess.

For microbiota, we mean the totality of single microorganisms—bacteria, fungi, archaea and protozoa—and of the viruses that live and colonize a specific environment in a given time. The human microbiota is defined as “the set of microorganisms that in a physiological, or sometimes pathological way, live in symbiosis with the human body”.

In recent years, the intestinal microbiota has been recognized as an important environmental factor in the pathophysiology of metabolic diseases. Through a molecular cross-talk with the host, it contributes to the maintenance of energy homeostasis and to the stimulation of immunity [51,63]. A causal link between intestinal microflora and host metabolism was first provided by Turnbaugh et al.; their study demonstrated that transplantation of intestinal microflora from obese mice could replicate the obese phenotype in previously germ-free mice [63].

The human gut is populated by different species of bacteria that co-evolve with the host from birth and maintain dynamic interactions with the human organism throughout life. Humans could be considered superorganisms whose metabolism represents the combination of microbiota and human characteristics [64]. The metabolic role of the intestinal microbiota is essential to human biochemical activity; it allows for the recovery of energy from otherwise indigestible foods and the production of vitamins and other essential nutrients [65]. Furthermore, the intestinal microbiota also regulates many aspects of innate and acquired immunity, protecting the host from the invasion of pathogenic germs and chronic inflammation. In contrast, imbalances in the composition of the gut microbiota, termed dysbiosis, have been associated with susceptibility to infections, immune alterations and, recently, insulin resistance and weight gain [66]. The intestinal microbiota will be then considered a separate organ that is involved, through a continuous molecular cross-talk with the host, in the maintenance of energy homeostasis and in the stimulation of immunity.

The human gastrointestinal tract is estimated to be colonized by over 100 trillion (10^14^) microbes, an estimated ten times more than the number of human cells, from over 1000 different species [67]. The composition of the gut microbiota is determined and influenced by several factors, such as genetics, age, geographic origin, diet and the use of different pharmacological agents, especially antibiotics [68,69].

Metagenomic analyses in lean mice and lean human volunteers have shown that almost all bacteria present in the distal intestine and feces belong to two main phyla, Bacteroidetes and Firmicutes, and most studies show a predominance of Bacteroidetes over Firmicutes [63,70], although this is not always uniformly proved [71,72].

Specifically, the distribution of the subclasses of the human intestinal microbiota consists of Bacteroidetes [23%], which include over 20 genera of Bacteroidetes, with Bacteroidales being the best known, in particular, the genus Bacteroides; Firmicutes [64%], which are Gram-positive bacteria, divided into various classes: Clostridia (the class of bacteria for which oxygen is toxic), Mollicutes (a class of tiny, obligate intracellular parasitic bacteria without cell wall], Bacilli [including bacteria such as Bacillus, Listeria, Staphylococcus) and Lactobacilli (a class of lactic bacteria); Proteobacteria [8%], which are Gram-negative bacteria, such as Escherichia coli and Helicobacter pylori; and Fusobacteria, Verrucomicrobia and Actinobacteria [3%], which include some species such as Bifidobacterium [73].

However, most studies have shown, instead, that this proportion changes to a high metabolic state, with less Bacteroidetes and more Firmicutes both in genetically obese ob/ob mice, diet-induced obesity (DIO) mice and obese humans compared to the normal microbiota [51,71]. This change leads to an increase in the fermentation of indigestible foods and the production of short-chain fatty acids (SCFAs), in particular butyrate, acetate and propionate. Butyrate is the main energy substrate for cellular metabolism in the colon epithelium; acetate and propionate are instead used by the liver and act as substrates for hepatic lipogenesis and gluconeogenesis [51,70]. Furthermore, the intestinal microbiota can also model the state of the host’s metabolism by influencing the expression of genes that regulate the accumulation and expenditure of host energy. SCFAs can influence host susceptibility to T2DM through epigenetic regulation of gene expression by inhibiting histone deacetylase [74].

All these data suggest that differences in the metabolic capacity of the gut microbiota in a single individual can significantly contribute to the development of various metabolic alterations, from obesity and insulin resistance to type 2 diabetes and metabolic syndrome, at least under certain conditions [70,75,76].

At the intestinal level, as already mentioned, the bacterial populations of the microbiota allow, through fermentation phenomena, to degrade the otherwise digestible carbohydrates to produce short-chain fatty acids (SCFAs) that heighten the energetic availability and allow for a more balanced host immune response [77].

Notably, SCFA production comprises metabolic cooperation between the bacterial community, because no bacterial species alone could hydrolyze all types of nutrients [63].

The butyrate produced represents the main energy source for the colon epithelium, approximately 60–70% of their energy needs; in addition, butyrate may also play an important role in regulating the growth and differentiation of these cells [78]. On the other hand, acetate and propionate can become a substrate for gluconeogenesis [79]. SCFAs are also important for maintaining the efficacy of the epithelial barrier; butyrate increases the production of mucus and essential proteins, such as zonulin and occludin, for tight junctions, contributing to the efficiency of the intestinal barrier [65].

SCFAs also have anti-inflammatory effects, partly through the inhibition of NF-B and partly through the mediation of acetate-induced anti-inflammatory stimuli, which promotes an activation of the host’s immune cells by binding to the G protein-coupled receptors GPR43 and GPR41, which are expressed not only in human adipocytes but also in colon epithelial cells and peripheral blood mononuclear cells [80,81].

Moreover, SCFAs’ protective action (butyrate, in particular) is essential to equilibrate the energetic element, as previously mentioned, but also to intervene in the modulation of gastrointestinal peptide production, increasing the protective effects of the intestinal barrier [81].

The intestinal microbiota also plays an important role in the regulation of host immunity. A dysbiotic microbiome is a predisposing factor to different outcomes of diseases associated with overeating. Immune changes caused by obesity will contribute to the onset of an inflammatory environment and intestinal dysfunctions, which will then have consequences for metabolic homeostasis [82,83], in a self-proliferating circuit. The obesity-induced pro-inflammatory shift of immune cells is a condition that favors the release of pro-inflammatory cytokines (TNF and IFNγ) and goes hand in hand with a reduction in the release of protective cytokines (IL-10 and IL-22) and adipokine. Furthermore, it is also associated with a reduced expression of antimicrobial proteins, produced by enterocytes, such as regenerating islet-derived protein 3 gamma (Reg3γ). These proteins interact with Toll-like receptors (TLRs), a family of proteins that play a key role in the body’s defense, particularly in innate immunity. They are noncatalytic transmembrane receptors, mainly expressed on the membrane of macrophages and dendritic cells. They recognize typical structures of pathogens and microbes and, for this reason, they are part of the superfamily of “pattern recognition receptors” (PRRs). Once the pathogen has breached the host’s anatomical barriers (e.g., human skin or intestinal mucosa), it is recognized by TLRs that activate the immune responses of the sentinel cells. Through this bond they manage to inactivate Gram-bacteria by binding to peptidoglycans present on the outside of the bacterial surface, thus favoring a condition of dysbiosis [84]. The pro-inflammatory state also induces a reduction in the production of mucin and epithelial proteins, such as claudin and occludin, which are essential for the integrity of the tight junctions of the intestinal barrier. These changes could influence the performance of the intestinal barrier and associated components, favoring the permeability to toxic bacterial products present in the intestinal lumen, which could subsequently pour into the bloodstream [65,85,86].

The two most common phyla in the gut microbiota differ in clinical classification according to Gram staining: Firmicutes are Gram-positive and Bacteroidetes are Gram-negative bacteria. Gram-negative bacteria, as we just described, contain a lipopolysaccharide (LPS) on the outer membrane; it is a large molecule consisting of a lipid portion and a polysaccharide and is capable of eliciting strong immune responses, being a powerful activator of Toll-like receptor 4 (TLR4), a receptor of the Toll-like receptor group that is present in most of the immunocompetent cells and macrophages and which allows for the recognition of molecular profiles associated with pathogens (i.e., pathogen-associated molecular patterns (PAMPs)). The link between LPS and TLR4 activates an extensive cellular signaling pathway that stimulates the inflammatory response and the release of cytokines [87]. Several studies [51,88] have shown that circulating levels of LPS are found at low concentrations in healthy individuals but are elevated in obese rodents and humans, in a condition called metabolic endotoxemia. Various mechanisms have been proposed to explain the links between obesity and metabolic endotoxemia: the intake of excess fat triggers an increase in chylomicrons in the intestine in the postprandial phase, which facilitates the transport of LPS into circulation; a diet rich in fats can promote a condition of dysbiosis, which determines an increase in intestinal permeability and, therefore, a passage into circulation by bacterial products such as LPS [51,88].

The permeability of the intestinal mucosa is largely regulated by the ability with which epithelial cells adhere to each other. The alteration of intestinal permeability is probably due to the reduced expression of claudin and occludin, the main proteins that make up the tight junctions of enterocytes. These two proteins protrude on the outer face of the membranes and are joined together by noncovalent bonds forming a belt around the cell that allows for adhesion between the enterocytes and preventing diffusion towards the intercellular spaces of enzymes and substances that would cause the digestion of the intestinal cells themselves and of the matrix, causing the permeability of the epithelial barrier [70]. If the barrier mechanism is not functioning properly (i.e., intestinal permeability), intestinal contents can leak into the circulatory system. This leads to the passage of pathogens but also toxins and allergens, including LPS, along with other metabolic products, which can also affect the functions of distant organs [89]. Once systemic circulation is reached, LPS migrate to different organs, such as the liver, or adipose tissue, triggering an innate immune response. In particular, LPS binds to plasma LPS-binding protein (LBP), which activates the receptor protein (cluster of differentiation 14 (CD14)) present on the plasma membrane of macrophages as part of the innate immune system.

The complex thus generated binds to the Toll-like 4 receptor (TLR4) of macrophages, which triggers transduction signals that activate the expression of different genes that code for different inflammatory factors, such as the activator protein 1 (AP-1), a transcription factor that regulates gene expression in response to a variety of stimuli, including cytokines, growth factors, stressors and bacterial and viral infections, and nuclear factor κB (NF- κB). NF-kB is a nuclear transcription factor present in all cells that produce cytokines, growth factors, chemokines, adhesion molecules, receptors for all of the molecules just mentioned and acute phase proteins, both in normal conditions and in numerous diseases. Once activated, NF-kB controls directly or with the cooperation of other transcription factors the activity of over 100 genes that regulate numerous cellular processes of vital importance for inflammation and the immune response [88,90].

The toxic effects of pathogens on Gram-negative bacteria, which are members of the gut microbiota, causes an excessive LPS presence. This molecule induces an increase in the gut membrane’s permeability, determining its major presence in the blood stream, but it also leads to an important macrophages passage in the intestinal lumen. This pro-inflammatory condition is very frequently found in obese subjects [87,88]. Gut membrane alterations with LPS migration in the blood stream must thus be considered as a critical factor in the development and sustainment of the evolution of chronic inflammation, which also happens to be a leading factor in the establishment of insulin resistance and metabolic syndrome [91] 

Excessive LPS liberations might be considered a determining link between external and internal gut inflammation, both key conditions for developing obesity and metabolic syndrome [91].

## 3. Discussion

Obesity has been considered a chronic inflammatory condition, albeit of a low grade, for several decades, caused by an unbalanced and high-calorie diet; an increase in the inflammatory tone has important impacts on the intermediate metabolism. For example, the expansion and infiltration of pro-inflammatory immune cells is present in several metabolically active cells and tissues during the development of T2DM [92]. This pro-inflammatory environment has vast consequences for the body’s functions, as seen in the development of insulin resistance, beta cell dysfunction and nonalcoholic fatty liver disease (NAFLD). Fatty liver disease is characterized as an excessive accumulation of lipids in hepatocytes that includes simple fatty infiltration (a benign condition called fatty liver). The presence of metabolic syndrome increases the likelihood of a patient having nonalcoholic steatohepatitis (NASH) rather than simple steatosis [39]. The nutritional alterations, consequent to unbalanced diets, also strongly influence the composition of the intestinal microbiota, causing a state of dysbiosis. The role of the intestinal microbiota in these complex processes has only recently begun to be understood. To enable an optimal symbiotic relationship between the human host and the gut microbiota, a controlled and adequate immune response is essential [93]. The human host can influence the microbiota through the diet. Eating a diet rich in fiber, for example, promotes the production of SCFAs which are able to improve energy homeostasis, glucose tolerance and, consequently, the regulation of an adequate inflammatory response. Both functions, those of the gut microbiota and those of the immune system, are interconnected and are dysfunctional in metabolic diseases [39].

The complex interaction involving the diet, the gut microbiota and the human host has been investigated for over a century now. Considering that the digestive tract constitutes the largest surface of the human body, with an extension between 30 and 40 m^2^ in adults and that hosts a remarkable and composite microbial community that lives in reciprocal and dynamic relationships with the human host. The intestinal microbiota is therefore a very complex ecosystem, as it hosts large populations of bacteria in the intestine and colon, with approximately 10^12^–10^1^⁴ microorganisms/gram, in close cooperation with each other [73,94].

Nutrition plays an important role in determining the quality and quantity of the gut microbiota and, of course, it varies from individual to individual. The composition of the intestinal microbiota may also be due to t altered microbiotic colonization, secondary to unhealthy family or social eating habits, during the first years of life, which can consequently determine the ability to collect energy from the diet even in subsequent years [95]. Long-term changes in the gut microbiota, such as lower levels of Bifidobacteria and higher levels of Bacteroides, have also been observed in children exposed to antibiotics during early childhood. Exposure to antibiotics, such as norfloxacin and ampicillin, can promote intestinal dysbiosis that can alter the hormonal, inflammatory and metabolic environment of the host [96]. These antibiotic-induced changes can predispose children to overweight and obesity [3].

Furthermore, the microbiome can also represent a fingerprint both of the hereditary genetic material of the human host and of the intestinal microbial environment. Indeed, it has been proposed that the microbiota gene pool represents an extension of the nuclear and mitochondrial genome, leading to the definition of metagenome to describe such an extension [73]. 

The intestinal microbiota is, thus, able to influence the functioning of both different organs and distal systems—resembling a new endocrine “virtual organ” [97]. 

### 3.1. Therapeutic Potential of Remodeling the Microbiotic Profile

The remodeling of bacterial strains in the digestive tract can help to positively configure the metabolic profile in an obese human host, as suggested by many data from animal and human studies.

Metabolic and signaling interactions between the immune system, the gut microbiota and the host have raised the concept of the therapeutic manipulation of the microbiome to counter or prevent obesity. In particular, the selection of specific intestinal bacterial strains and the enhancement of the intestinal bacterial flora may represent a promising therapeutic approach to control energy intake and reduce the prevalence of obesity and metabolic syndrome [51,98]. 

### 3.2. Remodeling of the Microbiota Secondary to Bariatric Surgery

In contrast to calorie restriction alone and exercise, bariatric surgery is considered an effective treatment for substantial and persistent weight loss in severely obese patients. Bariatric surgery restructures the anatomy of the intestine and, therefore, interferes with the feeding process; until recently, it was believed that weight loss was linked to the reduction of stomach and intestinal surfaces by altering the absorption of food [94]. Several recent studies suggest that the effectiveness of bariatric surgery is largely due to the fact of its effects on the intestinal microbiota [99,100,101]. Obese patients who underwent gastric bypass showed an increase in the Firmicutes-to-Bacteroidetes ratio that approached the microbial profile of lean subjects (as measured by the Shannon index) [102]. Some evidence reports that the physiological changes observed after bariatric surgery depended on an improvement in the microbiota. Mice colonized with microbiota from patients treated with a gastric bypass (RYBG) or vertical sleeve gastrectomy surgery (VSG) maintained a lower weight than those colonized with the obese microbiota collected before the surgical procedure [103,104]. A comparative study of three obese subjects, three thin and three after gastric bypass, revealed an increase in the percentage of Gammaproteobacteria (mainly Enterobacteriaceae) and Fusobacteriaceae, after surgery, accompanied by a proportional reduction in the levels of Firmicutes (i.e., Clostridium bacteria) and methanogens [105]. The authors hypothesized that the bypass of the upper small intestine can lead to the transfer of some bacteria typical of this tract (e.g., Enterobacteriaceae) into the large intestine; this determines a modification of the intestinal microenvironment, with consequent changes in the ingestion and digestion of food. In a larger study, 30 obese subjects enrolled in a follow-up bariatric surgery program and 13 lean control subjects were evaluated using a quantitative PCR-based fecal microbiota assay (qPCR), a method that simultaneously amplifies and quantifies the DNA of bacteria [106]. Before RYBG, obese patients showed a standard increase in the Firmicutes-to-Bacterioidetes ratio as well as a subsequent decrease at three and at six months after surgery, correlating directly with patients’ weight loss. An evident correlation has been reported between the levels of F. prausnitzii, E. coli and Bacteroides/Prevotella and the metabolic and inflammatory indices. The concentration of F. prausnitzii was negatively correlated with the serum concentrations of circulating inflammatory markers (hs-CRP and IL-6). Interestingly, leptin levels decreased after RYGB, while the E. coli concentration increased significantly [73].

### 3.3. Remodulation of the Microbiota-Fecal Microbiota Transplant (FMT) and Gut Microbiota Transplant (GMT)

Although bariatric surgery is considered an effective means in cases of high obesity, it is still an invasive procedure with possible associated risks. Therefore, the possibility of acting on the microbiota in a less invasive way remains a preferred option, particularly in moderate obesity and in the presence of associated symptoms.

Fecal microbiota transplant (FMT) is an efficient way to reshape the gut microbial ecosystem. In recent years, it has been shown to be useful in the treatment of C. difficile infections (CDIs), which are resistant to other therapies, achieving an 80–90% success rate in patients of different ages [107,108]. In addition, it can be used as a therapy for inflammatory bowel diseases [109], and this has led to hypothesizing the possibility of manipulating the microbiota of people suffering from obesity through FMT [110]. It was shown that in obese patients, heterologous microbiota FMT from nonobese people induced an improvement in insulin sensitivity, with a 2.5-fold increase in n-butyrate production by intestinal microbes such as R. intestinalis. The persistence of this positive effect for a minimum of 6 weeks after FMT supports the idea that gut microbiota transplantation (GMT) could offer an alternative to bariatric surgery [111]. In a study conducted on mice transplanted with feces with healthy microbiota and with the addition of ω3, there was a lower tendency to weight gain compared to the controls, despite eating a high-calorie diet [112]. At the moment, however, there is only evidence from mouse models of the ability to manipulate the microbiota by GMT in order to hinder obesity [94]. Currently, the use of FMT or GMT is only authorized as a treatment for C. difficile infections (CDIs), because there is still no unanimous consensus among researchers on the definition of optimal donor microbiota and on any operational procedures [94].

### 3.4. Remodeling of the Microbiota through Prebiotics and Probiotics

Supplementation with probiotics and prebiotics can help to remodel the microbiota in a positive way. For several years, the use of probiotics and prebiotics has been studied in subjects suffering from obesity and other metabolic diseases to improve the interactions between the intestinal microbial ecosystem and the metabolism of the host. Probiotics are living and active microorganisms (especially bacteria) that, if provided in sufficient numbers through supplements or food, can have a positive effect on health, in particular by strengthening the intestinal ecosystem and improving the balance of the intestinal microbiota. Specifically, in mice fed a high-fat diet, bacterial species such as Bifidobacterium spp. have been shown to improve glucose homeostasis and reduce weight gain and fat mass, as well as restore glucose-mediated insulin secretion [113]. The administration of probiotics in mice promoted the growth of Roseburia, which favored a decrease in blood glucose values, and this reduction may be the basis of the observed effects on weight loss and slowing of the progression towards T2DM [114,115]. 

Prebiotics are nondigestible substances naturally contained in some foods, mainly water-soluble, nongelling fibers, including nonstarch polysaccharides or beta-glucans, fructans, oligofructosaccharides (FOS), inulins (long-chain fructosyl-oligosaccharides), galacto-oligosaccharides (GOS), lactitol, lactosaccharose, lactulose, pyrodextrins and soy oligosaccharides, which are transformed by the intestinal microbiota into SCFAs and simultaneously promote the growth, in the colon, of one or more bacterial species useful for the development of probiotic microflora [51]. For example, inulin has been found to stimulate the growth of bifidobacterial. In animals, it may also reduce caloric intake and fat mass [116]. Additionally, the growth of bifidobacteria also correlates with increased glucose tolerance, improved glucose-induced insulin secretion and a tendency to normalize inflammation in rodents [117,118]. GOS can also modulate the absorption of monosaccharides from the intestine by modifying the activity of the monosaccharide transporters, which in turn induce the activation of glycolytic pathways [119]. In rodents, the consumption of prebiotics is also associated with reduced levels of lipids in plasma, liver and kidneys. In particular, the integration of GOS in the diet of healthy mice resulted in a reduction of triglycerides in the liver, reducing the activity of lipogenic enzymes and the synthesis of fatty acids and microsomal proteins involved in the synthesis of VLDL. The evidence suggests that the intake of prebiotics could reduce lipogenic activity and increase lipolytic activity [119,120].

Several studies in rodents have found that prebiotics and probiotics are effective not only in moderating weight gain and glucose metabolism but also in stimulating anti-inflammatory activity, and this activity is mainly due to the increase in SCFA production. SCFAs interact with the GPCRs (e.g., GPR41 and GPR43) of the immune cells in the colonic epithelium by promoting the expression of specific chemokines, reducing the action of NF-κB and the production of pro-inflammatory markers, such as IL-2 and IL-10, in leukocytes [51,121,122]. Other animal studies have found that SCFAs, in addition to their actions on host metabolism, have the ability to increase the sensation of satiety by increasing the synthesis of peptide YY (PYY) and proglucagon, a proenzyme of the protein hormone glucagon, in epithelial cells and inhibit the expression of some neuroendocrine factors, such as leptin [116,123]. However, the evidence for the anti-obesity effects of prebiotics still remains largely confined to animal model evaluations. In humans, studies currently show that the consumption of prebiotics have moderate to no impact on weight loss [51].

Prebiotics are present in many lactic ferment supplements but also in various foods, especially in wheat flour, bananas, honey, wheat germ, garlic, onion, beans and leeks.

The effect of prebiotics and probiotics in patients suffering from obesity and metabolic diseases, in humans, requires further investigation and numerically larger samples. In particular, it will be necessary to design studies that use specific strains, in appropriate doses of probiotics or prebiotics, together with controlled diets in order to evaluate the specific individual responses to the different types of interventions and to compare the impact on the intestinal microbiota with respect to the different possible variants, such as genetic or environmental factors [51].

### 3.5. Remodeling of the Microbiota through Physical Exercise

Intestinal microbial plasticity is impacted by many environmental factors, and among these physical exercise plays a significant role. In several studies on healthy animals, physical activity was found to affect the taxonomic composition of the microbiota [124,125]. However, not all research agrees on the changes that occur in the Firmicutes-to-Bacteroidetes ratio, finding in some cases a significant increase [126] and in others a clear reduction [127] or no significant change [128]. Recent research conducted on elite athletes [129] and on sedentary women who practice minimal physical activity [130] reported that, similar to animal models, physical exercise can shape the intestinal microbiota, favoring the reproduction of bacterial species useful to the body, such as the Prevotella genera, Copococcus (a producer of butyrate and protection factor in irritable bowel) and Bifidobacterium and the species F. prausnitzii, R. homini and A. muciniphila. The athletes also showed positive changes in the ratio of Firmicutes to Bacteriodetes. The effects of physical exercise, however, were transient, reversible and also influenced by multiple factors, including diet, age (the taxonomic composition varied during life), body composition (lean mass vs. fat mass assessed by BIA) and the type and extent of exercise (low vs. high intensity) [94].

## 4. Conclusions

The prevalence of obesity has increased in pandemic proportions in adults and children across much of the planet. Several factors have been identified to explain the etiology and pathogenesis of obesity, including diet, lifestyle, environmental factors and individual genetic factors. However, none of these fully explain the etiology of obesity and the search for possible causes continues [3]. For several years now, numerous studies have reported a significant association between obesity and metabolic dysfunctions with an inflammatory state and a peculiar composition of the intestinal microbiota. There are many mechanisms involved that can explain how the intestinal microbiota enters the pathophysiology of obesity. The intestinal microbial ecosystem interacts with the host’s metabolism on several levels. It collaborates to convert the complex nutrients ingested into SCFAs and transforms mucins and dietary fibers into simple sugars ready for absorption. It also stimulates intestinal epithelial proliferation, promotes the absorption and metabolism of nutrients and is the main actor in the remodeling of the intestinal barrier, a crucial structure that represents the borderline between the body and the external environment. In particular, it acts as an active filter and allows for control of the local immune system and connects with the systemic one [131]. The intestinal microbial populations also have a control action on the intake and expenditure of energy through the entero–endocrine cells from the intestine. The entero–endocrine cells in the gut respond to nutrient intake by secreting incretins and hormones such as glucagon-like peptide 1 and 2 (GLP-1 and GLP-2). GLP-1 stimulates the release of insulin from the pancreas, slows gastric emptying, increases the sense of satiety in response to food intake and reduces appetite by directly acting on the central nervous system’s hunger regulation centers and weight loss; while GLP-2, a peptide of 33 amino acids, co-secreted with GLP-1, mainly by intestinal L cells following food ingestion, improves intestinal glucose transport and plays an anti-inflammatory role. To date, it is known that this hormone plays a critical role in the trophism of intestinal crypts, stimulating their proliferation and inhibiting their apoptosis and reducing intestinal permeability [73,132,133].

There is no doubt that obesity and metabolic syndrome are associated with a state of chronic low-grade inflammation, as demonstrated by many data in the literature, and it is very likely that the initial trigger of metabolic inflammation is the interruption of energy homeostasis, produced by a positive energy balance, and that the induced inflammatory state is the initial adaptive response, designed to relieve the anabolic pressure produced by overeating. However, over time, this adaptive response can manifest itself as a maladaptive excess, indicating a failure to resolve as the initial insult perseveres [35]. Alterations in the intestinal microbiotic ecosystem can trigger or promote an inflammatory condition, with an increase in pro-inflammatory cytokines, a key factor in the development of obesity and metabolic syndrome. A condition of intestinal dysbiosis favors an alteration of the intestinal barrier; this kind of occurrence allows for transmigration outside of the intestinal lumen of a lipopolysaccharide (LPS), a large molecule present on the cell wall of Gram-negative bacteria such as Bacteroidetes, inducing a powerful inflammatory response, the deposition of adipose tissue and insulin resistance. Cani et al. were the first to indicate that bacterial LPS plays a key role in metabolic diseases related to a high-fat diet [91].

The microbiota–host interaction is in a constant dynamic arrangement, and the subject’s response can change adaptively from time to time. Attempts to intentionally modify the microbiota, at least in the short term, are possible by changing diet and lifestyle or by administering prebiotics, probiotics and antibiotics. Even the very effectiveness of bariatric surgery in treating obesity may be largely due to the fact of its effects on the microbiota, rather than simple anatomical remodeling, just as FMT helps to remodel gut microbial communities [94].

The use of prebiotics and/or probiotics could have a rationale in the attempt to counteract the development of excess weight through three main mechanisms of action. First, it has an antagonistic effect on the growth of pathogenic microorganisms and the competitive adherence to the intestinal mucosa and to the epithelium (i.e., antimicrobial activity). Secondly, it increases intestinal mucus production and decreases intestinal permeability (i.e., barrier function). Lastly, it modulates the gastrointestinal immune system (i.e., immunomodulation) [73,134].

Although several randomized trials on probiotics in obesity have been carried out, their results are still not convincing. Unfortunately, there is still no clear evidence on the efficacy of treatment with pre- or probiotics for obesity and metabolic syndrome. The articulated interaction between the complex microbial ecosystem, the inflammatory system and the intestinal system as a whole make it difficult to understand the regulatory mechanism. For example, the role of the virome [135] or mycobiome [136,137,138] is still unclear and poorly explored, and research is needed in this regard.

The variability of the therapeutic results can also be attributed to the heterogeneity of the experimental designs used for the different studies, such as the differences in the methodology of analysis of the fecal material, in the control of the diet, in the individual genetic susceptibility and to the different factors related to lifestyles. Furthermore, fecal sample analysis is the only reference for gut microbiota assessment and may not represent the true picture of the colon’s microbial population [3]. In the near future, further studies on the profiling and remodeling of microbiota in the treatment of obesity will be necessary, with more standardized analysis and comparison procedures, also using systems biology approaches, in order to confirm today’s promising results.

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
