# Peer review of "Obesity and Its Multiple Clinical Implications between Inflammatory States and Gut Microbiotic Alterations"

_diseases, 2022, doi:10.3390/diseases11010007_

Round 1

Reviewer 1 Report

This review summarizes the major evidences on interactions between the gut microbiota, energetic metabolism and the host immune system. But there are some questions in the aspects of experimental designs, results and discussion and so on. Hence, I have some suggestions as follows:

1) Some descriptions in the manuscript were not exact or confusing. Some words which will make the manuscript feel like an article on a popular science book should not appear in such a research paper. The following are suggestions for improving English usage. Please use standard expression in English.

2) Please add the analysis to every point in your Figure 2.

3) The manuscript stays within a stage of literature survey, and is hard to find original contribution of the authors on this subject.

 4) Problems on format or details: the manuscript was not well prepared according to the “Guidelines”. Please check carefully.

5) You had better transfer these general descriptions to special quantitative research.

Reviewer 2 Report

The manuscript is interesting. However, this reviewer believes it would benefit from some changes outlined in the document attached

Round 2

Reviewer 1 Report

can be published.

Reviewer 2 Report

The manuscript has only slightly improved.

This reviewer actually expected a major revision of the manuscript draft. Instead, the authors have provided minor changes. It is also not comprehensively elaborated (e.g. Arifuzzaman et al Narure 2022)

The manuscript has little to add to the body of literature in its current form.